# Sparse Disentangled VAE for Treatment Effect Estimation with Irrelevant Variables

## Abstract

Treatment effect estimation from imbalanced observational data is challenging, requiring balanced latent representations to reduce selection bias and enable accurate causal estimates. Many state-of-the-art methods employ VAEs with predetermined latent dimensionality, but this often causes over- or underfitting since too little relevant or too much irrelevant information is encoded. As cross-validating latent dimensionality is impractical for complex models and high-dimensional data, automatic determination is needed. We address this by learning sparsity-inducing masks that sub-select dimensions for each task, using a differentiable $L_0$ objective to penalize active dimensions and a mutual exclusivity regularizer to prevent overlap, ensuring independent and disentangled representations. Conflicting goals of accuracy and sparsity are balanced via Generalized ELBO with Constrained Optimization (GECO), optimizing sparsity only once prediction quality exceeds a threshold. Our method thus infers task-relevant latent factors, yields compact representations, and isolates irrelevant variables in challenging high-dimensional data. Experiments on real-world and synthetic datasets demonstrate improved predictive accuracy, compactness, and disentanglement compared to state-of-the-art baselines.

## 1 Introduction

Treatment effect estimation addresses causal reasoning questions such as: *What would have been the impact of a drug had an alternative treatment been administered?* It is a challenging task, and modern deep neural network methods involve learning latent factor representations that are essential for mitigating selection bias—non-random treatment assignment due to covariates—and ensuring robust, accurate estimates. Traditional propensity score-based methods to remove selection bias (Rosenbaum & Rubin., 1983; Rosenbaum, 1987; Li et al., 2016) often struggle, particularly in high-dimensional feature spaces where complex covariate–outcome relationships pose major challenges.

Therefore, deep neural network-based approaches (Johansson et al., 2016; Shalit et al., 2017; Yao et al., 2018; Hassanpour & Greiner, 2019b) have emerged as a compelling choice, especially in high-dimensional and large data settings. These methods learn balanced intermediate representations, enabling more precise predictions for the downstream prediction tasks associated with treatment effect estimation. Specifically, VAE-based approaches (Louizos et al., 2017; Zhang et al., 2021; Vowels et al., 2021) have been used to learn smooth and disentangled representations at the bottleneck. These methods utilize separate inference networks and optimize a joint objective consisting of reconstruction, treatment prediction and outcome prediction which—if balanced correctly—together can minimize selection bias in the learned latent representation.

However, a key issue in existing VAE-based approaches is predefined bottleneck width, i.e. the number of dimensions for representation of the latent factors. An overly large width can lead to overfitting, higher reconstruction error and spurious correlations between latent variables, while an insufficient width risks underfitting. Both cases undermine the accuracy of the target task (Bonheme & Grzes, 2023). Although cross-validation is a widely used solution, it is computationally expensive and impractical for large datasets. This issue becomes even more pronounced in large models with multiple encoders and irrelevant variables, where each encoder is tasked with representing a distinct latent factor. While the presence of irrelevant variables in large-scale observational data is inevitable, information leakage, redundancy and entanglement between distinct latent causal factors

interferes with the model's causal structure. In treatment effect estimation, suboptimal bottleneck width, irrelevant variables and information leakage between latent factors can lead to inaccurate inference of latent factors, resulting in spurious treatment effect estimates. We showcase this issue in our results in Tables 1a and 1b and Figures 2 and 4, where competing methods yield higher errors and entangled representations.

In this paper, we propose GLOVE-ITE (GECO and $L_0$ Optimization for Variational Estimation of Individual Treatment Effects), a novel VAE-based framework that estimates treatment effects while automatically determining the dimensionality of the latent representation, thereby eliminating the need for predefined or manually tuned bottleneck width. Furthermore, GLOVE-ITE also separates irrelevant factors into a distinct latent subspace, and keeps the representation of different causal factors separate to avoid redundancy and information leakage. To achieve this, we use a differentiable $L_0$ objective on the VAE bottleneck with a sparsity-inducing mask (Louizos et al., 2018). However, enforcing sparsity alone without outcome prediction constraint as part of the joint objective is detrimental as shown in our results (Figure 2). For this reason, we use Generalized ELBO with Constrained Optimization (GECO) (Rezende & Viola, 2018; Boom et al., 2020), which allows us to prioritize the prediction task over sparsity. We address the challenge of irrelevant variables with an additional, dedicated mask that separates irrelevant factors into a distinct latent subspace. Finally, we propose a mutual exclusivity regularization across all masks that prevents information leakage, yielding independent and disentangled representations, which is not achievable with traditional VAEs (see Figure 4). To the best of our knowledge, no existing method combines $L_0$ regularization with GECO to optimize the VAE bottleneck for treatment effect estimation, and more importantly, we address the overlap and information leakage between latent representations through a novel exclusivity regularizer.

Our approach demonstrates superior performance compared to state-of-the-art VAE-based methods on the two real-world datasets (IHDP, Jobs)(Brooks-Gunn et al., 1992; Hill, 2011; LaLonde, 1986; Dehejia & Wahba, 1999) as well as a challenging synthetic dataset (Hassanpour & Greiner, 2019a; Khan et al., 2024). We further provide valuable qualitative insights into our framework, enhancing its practicality and stability for real-world applications.

The core contributions of our work are:

- We propose a novel VAE-based framework that integrates GECO and $L_0$ regularization to automatically learn sparsity masks for each latent factor, while enforcing an outcome-prediction constraint to balance accuracy and sparsity, improving robustness in treatment effect estimation.
- We address irrelevant variables through a learnable mask and a dedicated latent subspace, yielding more accurate treatment effect estimation.
- We design a shared, compact encoder with an exclusivity regularizer that enforces independent usage of latent dimensions, enabling efficient learning of distinct factor representations and stronger disentanglement. Our method is validated via extensive qualitative analysis on real-world and synthetic datasets.

## 2 RELATED WORK

Selection bias, which arises due to non-random treatment assignment, is a well-known challenge in treatment effect estimation (Rosenbaum & Rubin., 1983), traditionally addressed through propensity score-based techniques such as matching, stratification, and re-weighting (Rosenbaum & Rubin., 1983; Rosenbaum, 1987; Li et al., 2016). However, these methods have limited effectiveness in high-dimensional real-world scenarios (Kuang et al., 2019). For high-dimensional settings, deep representation learning approaches have emerged as a powerful alternative for balancing individual samples (Johansson et al., 2016; Shalit et al., 2017; Yao et al., 2018; Hassanpour & Greiner, 2019b). These methods aim to mimic a Randomized Controlled Trial (RCT) by reducing bias between treatment groups in a joint embedding space for all variables. However, they fail to consider that not all covariates typically contribute to both the treatment and the outcome.

In contrast, disentanglement approaches further advance this idea by additionally separating instrumental, confounding, and adjustment latent factors to better capture underlying causal structures and have proved effective in addressing selection bias (Kuang et al., 2017; Hassanpour & Greiner,

2019a; Kuang et al., 2020; Wu et al., 2023; Cheng et al., 2022). Here, the latent space is often learned with variational autoencoders, e.g. focussing solely on confounding factors (Louizos et al., 2017) or separating all three latent factors (Zhang et al., 2021).

However, separating out irrelevant factors is highly relevant for treatment effect estimation because they can lead to spurious effect estimation and it was observed that higher number of irrelevant variables make state-of-the-art approaches fail (Khan et al., 2024). Therefore, models with four separate encoders and suitable loss functions have been suggested to deal with irrelevant factors (Vowels et al., 2021; Khan et al., 2024).

Despite their strengths, deep disentanglement and VAE-based approaches are sensitive to the number of latent factor dimensions, making determining the right latent dimensionality crucial to performance and avoiding spurious treatment effect estimates. Existing methods for this often rely on the elbow method or expensively training multiple models (Mai Ngoc & Hwang, 2020; Doersch, 2021; Bonheme & Grzes, 2022).

In contrast to tedious and expensive search for suitable bottleneck dimensionality, we propose a novel VAE-based solution leveraging a $L_0$ sparsity and constrained optimization objective. This approach is connected to intrinsic dimension estimation, which identifies effective latent size needed to describe data and has been explored for other deep learning tasks (Levina & Bickel, 2004; Facco et al., 2017; Gong et al., 2018; Ansuini et al., 2019; Pope et al., 2021).

Finally, the way Boom et al. (2020) induces sparsities in VAE bottlenecks unrelated to the treatment effect estimation problem serves as an inspiration for this work. Conversely, we address the problem by learning multiple masks from a shared latent space with disentangled representations and our $L_0$ sparsity objective is computationally simpler and avoids multiple forward passes per data point.

## 3 PROBLEM FORMULATION

An observational dataset $\mathcal{D} = \{\mathbf{x}_i, t_i, y_i\}_{i=1}^N$ consists of pre-treatment variables $\mathbf{x}_i \in \mathcal{X} \subseteq \mathbb{R}^K$, binary treatments $t_i \in \mathcal{T}$ (e.g., 0: medication, 1: surgery), and real-valued observed outcomes $y_i \in \mathcal{Y} \subseteq \mathbb{R}$ (e.g., recovery time). However, only the factual outcomes $y_i^{t_i}$ are observed, while the counterfactual outcomes $y_i^{1-t_i}$ remain unobserved. Selection bias is present when treatment assignment depends on $\mathbf{x}_i$, violating the randomized controlled trial assumption $P(\mathcal{T} \mid \mathcal{X}) = P(\mathcal{T})$. The objective is to estimate the Individual Treatment Effect (ITE), defined as $\delta_i = y_i^1 - y_i^0$, by learning a function $f \colon \mathcal{X} \times \mathcal{T} \to \mathcal{Y}$ (Hassanpour & Greiner, 2019a).

$$\widehat{\delta}_i = \begin{cases} y_i^1 - f(\mathbf{x}_i, 1 - t_i), & \text{if } t_i = 1 \\ f(\mathbf{x}_i, 1 - t_i) - y_i^0, & \text{if } t_i = 0 \end{cases} \tag{1}$$

To mitigate selection bias for reliable treatment effect estimates, it is essential to infer the underlying entangled latent factors that generate $\mathcal{D}$ (Hassanpour & Greiner, 2019a): instrumental ($\Gamma$: affect treatment), confounding ($\Delta$: affect both treatment and outcome), and adjustment ($\Upsilon$: affect outcome only). In high-dimensional settings, irrelevant variables ($\Omega$) are common, and their entanglement with relevant factors induces spurious dependencies. This issue is amplified in large latent spaces, biasing representations and worsening treatment effect estimates if not explicitly addressed (Kuang et al., 2017; Khan et al., 2024), as shown in Table 1a. Further details on the underlying ITE assumptions and VAE background can be found in Appendix A.1.1 and A.1.2.

## 4 METHOD

Our approach builds on the VAE model as shown in Figure 1, where the observed pre-treatment variables $\mathbf{x}$ are encoded into a shared latent space $\mathbf{z} = \{\mathbf{z}_{\Gamma,\Delta,\Upsilon,\Omega}\}$. From this space, we derive task-specific masked representations $\mathbf{z}'_\Gamma$, $\mathbf{z}'_\Delta$, $\mathbf{z}'_\Upsilon$, and $\mathbf{z}'_\Omega$ (e.g., $\mathbf{z}'_\Gamma = \mathbf{z}_\Gamma \odot \mathbf{m}_\Gamma$), collectively denoted as $\mathbf{z}' = \{\mathbf{z}'_{\Gamma,\Delta,\Upsilon,\Omega}\}$. The standard VAE objective (equation 12) is extended with two components: (i) a treatment prediction term $\log p(t \mid \mathbf{z}'_{\Gamma,\Delta})$, implemented as Binary Cross-Entropy (BCE), guiding $\Gamma$ and $\Delta$; (ii) a constraint outcome prediction term $C\left(\hat{y}'_{\Delta,\Upsilon}, t\right)$ implemented as Mean Squared Error with constraint (MSEC), to learn $\Delta$ and $\Upsilon$; and the reconstruction term $\log p(\mathbf{x} \mid \mathbf{z}'_{\Gamma,\Delta,\Upsilon,\Omega})$, together with the exclusivity loss $\mathcal{L}_{excl}$, helps to learn $\Omega$. We denote $\hat{t}'_{\Gamma,\Delta}$ and $\hat{y}'_{\Delta,\Upsilon}$ as the predicted treatment and outcome, respectively.

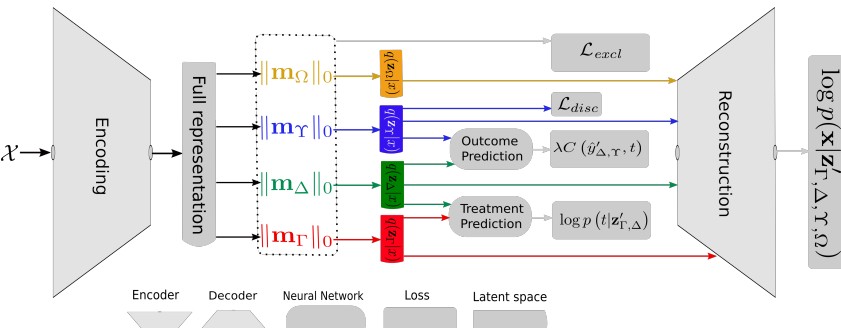

Figure 1: Architecture of GLOVE-ITE: Four latent subspaces are learned from a shared embedding via separate masks ($\|\mathbf{m}_\Gamma\|_0, \|\mathbf{m}_\Delta\|_0, \|\mathbf{m}_\Upsilon\|_0, \|\mathbf{m}_\Omega\|_0$). These subspaces serve distinct roles: $\Gamma, \Delta$ for treatment prediction ($\log p(t \mid \mathbf{z}'_{\Gamma,\Delta})$), $\Delta, \Upsilon$ for outcome prediction and constraint enforcement ($C(\hat{y}'_{\Delta,\Upsilon}, t)$), $\Upsilon$ for discrepancy loss ($\mathcal{L}_{disc}$), and $\Omega$ for irrelevant variables. The exclusivity loss ($\mathcal{L}_{excl}$) promotes disentanglement by discouraging overlap across latent dimensions. All subspaces jointly optimize the reconstruction objective ($\log p(\mathbf{x} \mid \mathbf{z}'_{\Gamma,\Delta,\Upsilon,\Omega})$).

Our objective is principled, extending the ELBO with an $L_0$ sparsity term (Louizos et al., 2018), GECO-based constraints (Rezende & Viola, 2018), and an exclusivity penalty. This formulation, detailed in equation 8 and equation 9, provides a novel and theoretically grounded approach for robust treatment effect estimation. Below, we first explain our approach to minimizing the number of active latent dimensions in $\mathbf{z}_{\Gamma,\Delta,\Upsilon,\Omega}$ with a differentiable $L_0$ objective (Section 4.1), then address the conflict between prediction and sparsity objectives using GECO (Section 4.2), enforce independence and disentanglement via mutual exclusivity loss (Section 4.3), and detail the overall objective implementation in Section 4.4.

## 4.1 $L_0$ SPARSITY OBJECTIVE

Our goal is to reduce the causal factors $\mathbf{z}_{\Gamma,\Delta,\Upsilon,\Omega}$ to their true dimensions for downstream prediction tasks to avoid bias and irrelevant factors. We introduce binary masks $\mathbf{m}_\Gamma$, $\mathbf{m}_\Delta$, $\mathbf{m}_\Upsilon$, and $\mathbf{m}_\Omega$ to define masked latent representations (e.g., $\mathbf{z}'_\Gamma = \mathbf{z}_\Gamma \odot \mathbf{m}_\Gamma$). While $L_1$ and $L_2$ regularizers are gradient-friendly and lead to small weights, the $L_0$ objective

$$\|\mathbf{m}\|_0 = \sum_{j=1}^{|\mathbf{m}|} \mathbb{I}[\mathbf{m}_j \neq 0]$$

induces actual sparsity by counting non-zero mask elements, yet is non-differentiable. To overcome this, we adopt a probabilistic relaxation: $\mathbf{m} \sim \text{Bernoulli}(\boldsymbol{\pi})$ with sparsity objective $\sum_{j=1}^{|\mathbf{m}|} \pi_j$, where $\boldsymbol{\pi}$ are activation probabilities (Louizos et al., 2018). Since $\mathbf{m}$ remains discrete, we apply the reparameterization trick with the Binary Concrete distribution (Maddison et al., 2017). Sampling $\mathbf{u}_i \sim \text{Uniform}(0, 1)$, we set

$$\mathbf{s}_i = \text{sigmoid}\left(\frac{\log \mathbf{u}_i - \log(1-\mathbf{u}_i) + \log \boldsymbol{\alpha}_i}{\beta}\right), \tag{2}$$

where $\boldsymbol{\alpha}_i$ and $\beta$ are location and temperature. As $\beta \to 0$, this converges to Bernoulli. We further obtain $\mathbf{m}$ by stretching the distribution with $\tilde{\mathbf{s}}_i = \mathbf{s}_i(\zeta - \gamma) + \gamma$ using $\gamma < 0$ and $\zeta > 1$. The differentiable sparsity objective becomes

$$\|\mathbf{m}\|_0 \approx \sum_{j=1}^{|\mathbf{m}|} \text{sigmoid}\left(\log \alpha_j - \beta \log\left(-\frac{\gamma}{\zeta}\right)\right). \tag{3}$$

However, combining this sparsity with reconstruction and prediction losses causes conflicts: overly strict sparsity hurts prediction, while weak sparsity retains redundant dimensions, increasing bias and reliance on irrelevant factors. It is not trivial to set good constant weights to balance the objectives as seen in Figure 2.

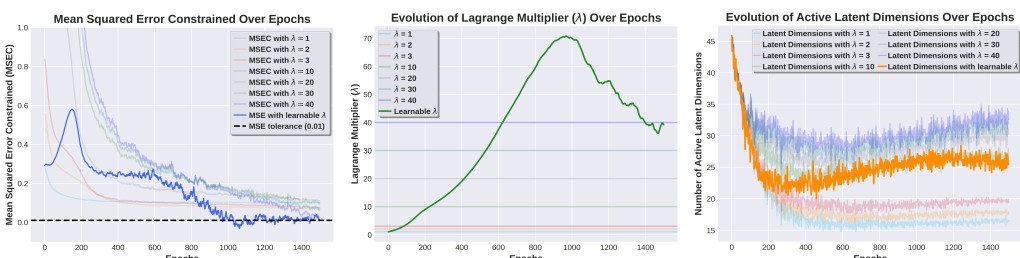

Figure 2: Visualization of three key training metrics: Mean Squared Error with constraint (MSEC:can be negative) relative to the threshold $\tau$, the Lagrange multiplier ($\lambda$), and the number of active latent dimensions on synthetic data with an 8×8×8 dimensional structure. Dimmed lines represent fixed $\lambda$, while brighter lines indicate learnable $\lambda$ (best viewed in color).

### 4.2 Prioritizing prediction performance over sparsity

During the training of the VAE model, the reconstruction and outcome prediction objectives naturally conflict with the sparsity objective from the previous Section (4.1). We address this problem by prioritization and only optimize sparsity as long as other tasks perform better than a threshold $\tau$. Generalized ELBO with Constrained Optimization (GECO) Rezende & Viola (2018) achieves this effect for standard VAEs by replacing the reconstruction loss with a constrained expression $C(\dots)$ and introducing a Lagrange multiplier $\lambda$ that is adapted during learning based on the constraint value. We adopt this approach for the outcome prediction loss:

$$\mathcal{L}_{\text{ELBO}} = \mathbb{E}_{q(\mathbf{z}_{\Gamma,\Delta,\Upsilon,\Omega}|\mathbf{x})}\Big[\dots + \lambda C\left(\hat{y}'_{\Delta,\Upsilon}, t\right)\Big] - D_{\text{KL}}(\dots), \tag{4}$$

where the predicted outcome $\hat{y}'_{\Delta,\Upsilon}$ only has access to the masked sections of the latent representation $\mathbf{z}'_{\Delta}$ and $\mathbf{z}'_{\Upsilon}$. This loss minimizes the KL divergence while enforcing the constraint $\mathbb{E}_{q(\mathbf{z}_{\Gamma,\Delta,\Upsilon,\Omega}|\mathbf{x})}\left[C\left(\hat{y}'_{\Delta,\Upsilon}, t\right)\right] \leq 0$. We define the constraint expression to enforce an upper bound on the masked outcome prediction task,

$$C\left(\hat{y}'_{\Delta,\Upsilon}, t\right) = \left\|y - \hat{y}'_{\Delta,\Upsilon}\right\|^2 - \tau, \tag{5}$$

where $y$ is the factual outcome (from the dataset) and $\hat{y}'_{\Delta,\Upsilon}$ is predicted from a regressor $h^t(y'_{\Delta,\Upsilon})$. The modified ELBO form equation 9 is optimized via a min-max optimization scheme (i.e. dual gradient decent), balancing the objectives and constraints effectively by adjusting $\lambda$ during training. The influence of this constraint and the value of $\lambda$ during training can be observed in Figure 2.

### 4.3 Mutual Exclusivity Regularization

While standard VAEs encourage disentanglement, learning multiple representations from a shared latent space can cause information leakage and duplication, leading to entangled representations shown in Figure 4. We introduce a principled softmax-entropy based exclusivity loss to smoothly penalizes overlap across representations. Let $\mathcal{M} \in \mathbb{R}^{4 \times d}$ be the stacked mask logits ($d$: latent dimensionality). A temperature-controlled softmax yields normalized assignments:

$$\mathbf{S} = \text{softmax}\left(\tfrac{\mathcal{M}}{\kappa}\right), \tag{6}$$

where $\kappa > 0$ controls assignment sharpness. The loss is the mean column-wise entropy of $\mathbf{S}$:

$$\mathcal{L}_{\text{excl}} = -\frac{1}{d}\sum_{j=1}^{d}\sum_{i=1}^{4} S_{i,j} \log S_{i,j}. \tag{7}$$

Minimizing $\mathcal{L}_{\text{excl}}$ drives each dimension toward a single mask, reducing redundancy and promoting disentanglement in latent representations.

### 4.4 Implementation Details

The overall loss combines: (i) the $L_0$ sparsity objective on masks $\mathbf{m}_\Gamma$, $\mathbf{m}_\Delta$, $\mathbf{m}_\Upsilon$, and $\mathbf{m}_\Omega$ (Section 4.1); (ii) the modified ELBO with outcome-prediction constraint (Section 4.2); (iii) the mutual

exclusivity loss (Section 4.3); and (iv) the discrepancy loss $\mathcal{L}_{disc}$ on latent representation across treatment groups.

$$\mathcal{L}_{GLOVE-ITE} = \mathcal{L}_{\text{ELBO}} + \|\mathbf{m}_{\Gamma,\Delta,\Upsilon,\Omega}\|_0 + \mathcal{L}_{excl} + \mathcal{L}_{disc}. \tag{8}$$

With all terms, the modified ELBO has the following form,

$$\begin{aligned}
\mathcal{L}_{\text{ELBO}} = \mathbb{E}_{q(\mathbf{z}_{\Gamma,\Delta,\Upsilon,\Omega}|\mathbf{x})}\Big[ &\log p\left(\mathbf{x}|\mathbf{z}'_{\Gamma,\Delta,\Upsilon,\Omega}\right) + \log p\left(t|\mathbf{z}'_{\Gamma,\Delta}\right) \\
&+ \lambda C\left(\hat{y}'_{\Delta,\Upsilon}, t\right)\Big] - D_{\text{KL}}(q(\mathbf{z}'_{\Gamma,\Delta,\Upsilon,\Omega}|\mathbf{x}) \,\|\, p(\mathbf{z}_{\Gamma,\Delta,\Upsilon,\Omega}))
\end{aligned} \tag{9}$$

Finally, the discrepancy loss is defined between the masked representations of adjustment latent factor ($\mathbf{z}'_\Upsilon$) under the two treatment assignments, i.e., $\mathbf{z}'_\Upsilon \mid t = 0$ and $\mathbf{z}'_\Upsilon \mid t = 1$.

$$\mathcal{L}_{disc} = wass[\mathbf{z}'_\Upsilon|t = 0, \mathbf{z}'_\Upsilon|t = 1], \tag{10}$$

$\mathcal{L}_{disc}$ ensures that $\Upsilon$ remains independent of $\Gamma$, effectively reducing the influence of $\Gamma$ on $\Upsilon$. This mitigates selection bias introduced by $\Gamma$, enabling unbiased predictions for the downstream task. To achieve this, we employ the Wasserstein distance as the discrepancy loss as proposed by (Shalit et al., 2017).

## 5 EXPERIMENTS

In this section, we present the evaluation criteria for treatment effect estimation, describe the datasets utilized in our experiments, experiment details, results and conclude with a qualitative analysis of the proposed method.

### 5.1 EVALUATION CRITERIA

A widely used metric for evaluating individual treatment effect estimation is the *Precision in Estimation of Heterogeneous Effect (PEHE)*:

$$PEHE = \sqrt{\frac{1}{N}\sum_{i=1}^{N}(\hat{e}_i - e_i)^2}, \tag{11}$$

where $\hat{e}_i = \hat{y}_i^1 - \hat{y}_i^0$ is the predicted effect and $e_i = y_i^1 - y_i^0$ the true effect. We also employ a policy risk ($\mathcal{R}_{pol}$) criterion for the Jobs dataset (details in Appendix B.2).

### 5.2 DATASETS AND EXPERIMENT DETAILS

We evaluate our approach on two real-world datasets and one synthetic dataset. Synthetic data enables qualitative analysis of the model while assessing its performance in treatment effect estimation.

Infant Health and Development Program (IHDP): The IHDP dataset, derived from an RCT by Brooks-Gunn et al. (1992) and adapted by Hill (2011) to introduce selection bias, contains 25 covariates describing child and mother characteristics. It includes 747 instances (139 treated, 608 control) and evaluates the effect of specialist home visits on children's cognitive health. We augment it with artificial contrasts to introduce irrelevant variables (Khan et al., 2024). Details of the Jobs dataset are provided in Appendix B.1.

Synthetic: We use the same settings of synthetic datasets as used by Hassanpour & Greiner (2019a); Khan et al. (2024), generated with a sample size $N$, dimensions $[d_\Gamma, d_\Delta, d_\Upsilon, d_\Omega]$, and specified mean and covariance matrices $(\mu_L, \sum_L)$ for each latent factor $L \in [\Gamma, \Delta, \Upsilon, \Omega]$. Data is sampled from a multivariate normal distribution, forming a covariate matrix of size $N \times (d_\Gamma + d_\Delta + d_\Upsilon + d_\Omega)$. Irrelevant variables are added by permuting values from other factors to ensure realistic complexity. Further details about network architectures, hyperparameters, and other experimental settings used for the experiments are provided in Appendix A.2 and B.3.

Table 1: PEHE (mean(std)) on IHDP (left) and synthetic (right) datasets with varying numbers of irrelevant variables ($\#\Omega$). Lower PEHE is better. Reported values also include the learned active latent dimensions per method.

| Method | $\#\Omega$ | $\Upsilon$-active | $\Delta$-active | $\Gamma$-active | $\Omega$-active | Total-active | PEHE |
|---|---|---|---|---|---|---|---|
| CEVAE | 5 | NA | 35 | NA | NA | 35 | 1.26 (0.13) |
| DR-CFR | 5 | 35 | 35 | 35 | NA | 105 | 1.09 (0.51) |
| TEDVAE | 5 | 35 | 35 | 35 | NA | 105 | 0.88 (0.62) |
| TVAE | 5 | 35 | 35 | 35 | 35 | 140 | 0.87 (0.47) |
| DRI-ITE | 5 | 35 | 35 | 35 | 35 | 140 | 1.09 (0.53) |
| **GLOVE-ITE** | 5 | 7 | 9 | 9 | 7 | 32 | **0.70 (0.07)** |
| CEVAE | 10 | NA | 35 | NA | NA | 35 | 1.54 (0.14) |
| DR-CFR | 10 | 35 | 35 | 35 | NA | 105 | 1.15 (0.61) |
| TEDVAE | 10 | 35 | 35 | 35 | NA | 105 | 1.11 (0.80) |
| TVAE | 10 | 35 | 35 | 35 | 35 | 140 | 1.10 (0.49) |
| DRI-ITE | 10 | 35 | 35 | 35 | 35 | 140 | 1.15 (0.51) |
| **GLOVE-ITE** | 10 | 7 | 9 | 9 | 8 | 33 | **0.74 (0.08)** |
| CEVAE | 15 | NA | 35 | NA | NA | 35 | 1.66 (0.18) |
| DR-CFR | 15 | 35 | 35 | 35 | NA | 105 | 1.19 (0.58) |
| TEDVAE | 15 | 35 | 35 | 35 | NA | 105 | 1.27 (0.87) |
| TVAE | 15 | 35 | 35 | 35 | 35 | 140 | 1.23 (0.52) |
| DRI-ITE | 15 | 35 | 35 | 35 | 35 | 140 | 1.18 (0.58) |
| **GLOVE-ITE** | 15 | 7 | 8 | 8 | 7 | 30 | **0.75 (0.08)** |

(a) IHDP dataset

| Method | $\#\Omega$ | $\Upsilon$-active | $\Delta$-active | $\Gamma$-active | $\Omega$-active | Total-active | PEHE |
|---|---|---|---|---|---|---|---|
| CEVAE | 5 | NA | 30 | NA | NA | 30 | 0.60 (0.019) |
| DR-CFR | 5 | 30 | 30 | 30 | NA | 90 | 0.23 (0.001) |
| TEDVAE | 5 | 30 | 30 | 30 | NA | 90 | 0.22 (0.012) |
| TVAE | 5 | 30 | 30 | 30 | 30 | 120 | 0.23 (0.003) |
| DRI-ITE | 5 | 30 | 30 | 30 | 30 | 120 | 0.26 (0.024) |
| **GLOVE-ITE** | 5 | 7 | 8 | 7 | 7 | 29 | **0.15 (0.005)** |
| CEVAE | 10 | NA | 30 | NA | NA | 30 | 0.60 (0.001) |
| DR-CFR | 10 | 30 | 30 | 30 | NA | 90 | 0.27 (0.016) |
| TEDVAE | 10 | 30 | 30 | 30 | NA | 90 | 0.50 (0.054) |
| TVAE | 10 | 30 | 30 | 30 | 30 | 120 | 0.28 (0.004) |
| DRI-ITE | 10 | 30 | 30 | 30 | 30 | 120 | 0.27 (0.027) |
| **GLOVE-ITE** | 10 | 7 | 7 | 7 | 7 | 28 | **0.16 (0.003)** |
| CEVAE | 15 | NA | 30 | NA | NA | 30 | 0.56 (0.005) |
| DR-CFR | 15 | 30 | 30 | 30 | NA | 90 | 0.28 (0.019) |
| TEDVAE | 15 | 30 | 30 | 30 | NA | 90 | 0.49 (0.010) |
| TVAE | 15 | 30 | 30 | 30 | 30 | 120 | 0.34 (0.003) |
| DRI-ITE | 15 | 30 | 30 | 30 | 30 | 120 | 0.28 (0.021) |
| **GLOVE-ITE** | 15 | 8 | 8 | 8 | 7 | 31 | **0.17 (0.005)** |

(b) Synthetic dataset

## 5.3 RESULTS

We compare against four state-of-the-art and VAE-based disentanglement methods that address irrelevant variables, ensuring methodological relevance and fair evaluation of latent compactness: CEVAE (Louizos et al., 2017), DR-CFR (Hassanpour & Greiner, 2019a), TEDVAE (Zhang et al., 2021), TVAE (Vowels et al., 2021), and DRI-ITE (Khan et al., 2024).

The Table 1a presents a comprehensive evaluation of treatment effect estimation methods on the IHDP dataset, highlighting the superior performance of our approach across varying dimensions of irrelevant variables ($\Omega$). GLOVE-ITE consistently achieves the lowest Precision in Estimation of Heterogeneous Effect (PEHE) values: 0.70 (0.07), 0.74 (0.08), and 0.75 (0.08) for $\Omega = 5, 10, 15$, respectively. Notably, GLOVE-ITE achieves this accuracy with significantly reduced total dimensionality (e.g., 32 vs. 140 for $\Omega = 5$), showcasing its ability to narrow down the VAE bottleneck and discard unnecessary latent dimensions. Furthermore, GLOVE-ITE demonstrates remarkable robustness to increasing irrelevant dimensions, exhibiting minimal degradation in PEHE compared to other methods. In contrast, baselines either rely on higher dimensional representations or exhibit inferior accuracy, underscoring the efficiency and effectiveness of our approach in solving complex treatment effect estimation tasks.

Table 1b highlights the effectiveness of various methods on a synthetic dataset structured with instrumental, confounding, adjustment, and irrelevant features ($8 \times 8 \times 8 \times \Omega$), focusing on the ability to estimate treatment effects while identifying and leveraging the true underlying data dimensions. GLOVE-ITE demonstrates its ability to learn optimal latent representations, achieving the lowest PEHE values: 0.15 (0.005), 0.16 (0.003), and 0.17 (0.005) for $\Omega = 5, 10, 15$, respectively. Importantly, it uses only 28–31 latent dimensions, closely aligning with the actual data regardless of $\Omega$, in contrast to baselines methods, which rely on significantly larger total dimensionalities (90–120). This dimensional efficiency underscores our method's ability to disentangle relevant factors from irrelevant ones while maintaining robust performance. As $\Omega$ increases, GLOVE-ITE effectively adapts to the true dimensions of the data, showing minimal degradation in PEHE compared to other methods. These results demonstrate the capability of our approach to optimize the use of latent dimensions in variational autoencoders, achieving state-of-the-art accuracy with an efficient and interpretable representation. On the Jobs dataset, GLOVE-ITE consistently outperforms competing VAE-based methods by achieving both lower policy risk and more compact representations (see Appendix B.4).

### 5.4 QUALITATIVE EVALUATION OF MODEL PERFORMANCE

Figure 2 illustrates the learning dynamics of our model, focusing on three key quantities: (1) Mean Squared Error with constraint (MSEC), (2) the Lagrange multiplier ($\lambda$), and (3) the number of active latent dimensions during training. To ensure stability and a reliable estimate, we show the average of the last five active dimension values after each epoch instead of the final value alone. Early in training, the $L_0$ regularizer aggressively reduces the number of active latent dimensions, causing an initial rise in MSEC around the $200^{th}$ epoch. Simultaneously, $\lambda$ increases steadily to enforce the MSE constraint. As training progresses, the MSEC gradually decreases toward the specified tolerance ($\tau$), while the number of active latent dimensions increases to balance dimensional efficiency with predictive accuracy. By the end of training, the MSEC stabilizes near $\tau$, the number of active dimensions converges, and $\lambda$ reaches a stable value, indicating successful convergence. Notably, models with fixed $\lambda$ fail to achieve comparable results, struggling to balance the trade-off between the two objectives effectively.

**Ablation study:** Table 2 presents the results of an ablation study on the loss function, evaluated on the IHDP dataset with five irrelevant variables. Starting with the baseline objective $\mathcal{L}_{\text{ELBO}} + \mathcal{L}_{disc}$, we observe a PEHE of 0.81 (0.39).

Table 2: Ablation study on IHDP (5 irrelevant variables). Bold indicates best PEHE (lower is better).

| Loss | PEHE |
|---|---|
| $\mathcal{L}_{\text{ELBO}} + \mathcal{L}_{disc}$ | 0.81(0.39) |
| $\mathcal{L}_{\text{ELBO}} + \|\mathbf{m}\|_0$ | 0.77(0.40) |
| $\mathcal{L}_{\text{ELBO}} + \|\mathbf{m}\|_0 + \mathcal{L}_{disc}$ | 0.75(0.39) |
| $\mathcal{L}_{\text{ELBO}} + \|\mathbf{m}\|_0 + \mathcal{L}_{disc} + \mathcal{L}_{excl}$ | **0.70(0.07)** |

Table 3: Compression analysis on synthetic 8×8×8×5 data under varying initial dimensionality. The learned representations consistently compress to ∼29–31 active dimensions, closely matching the true underlying size of 29.

| Initial | $\Gamma$ | $\Delta$ | $\Upsilon$ | $\Omega$ | Total |
|---|---|---|---|---|---|
| 60 | 8 | 9 | 8 | 6 | 31 |
| 68 | 7 | 8 | 8 | 5 | 28 |
| 76 | 5 | 9 | 9 | 6 | 29 |
| 84 | 9 | 7 | 8 | 5 | 29 |

Introducing the sparsity-inducing term $\|\mathbf{m}_{\Gamma,\Delta,\Upsilon,\Omega}\|_0$ reduces the error to 0.77 (0.40), highlighting the benefit of explicit dimension selection. Combining both the sparsity term and the discrepancy loss further improves performance to 0.75 (0.39). Finally, adding the exclusivity loss $\mathcal{L}_{excl}$ yields the best result with a PEHE of 0.70 (0.07), demonstrating that encouraging non-overlapping activation across masks enhances the model's ability to isolate relevant latent dimensions and improves treatment effect estimation accuracy.

**Compression analysis:** Table 3 presents a compression analysis of the proposed method on synthetic data with an 8×8×8×5 dimensional structure, evaluated across varying initial dimensionality settings. Despite starting with different initial dimensions (ranging from 60 to 84), the method consistently compresses the data to a compact latent representation, utilizing only 29–31 total dimensions, closely matching the true underlying size of 29. This highlights the model's ability to adaptively allocate dimensions among $\Gamma, \Delta, \Upsilon$ and $\Omega$ encoders, with minor variations depending on the initial latent dimensionality. For instance, with initial dimensions of 76 or 84, the method converges to 29 latent dimensions, while for 60 and 68, it stabilizes at 31 and 28 dimensions. These results emphasize the model's robustness in achieving efficient compression irrespective of the initial input dimensionality, while preserving the underlying data structure.

**Inference and Disentanglement:** Figure 3 highlights the inference of latent factors using the permutation feature importance theory (Fisher et al., 2018; Khan et al., 2024). Specifically, it demonstrates that only $\Gamma$

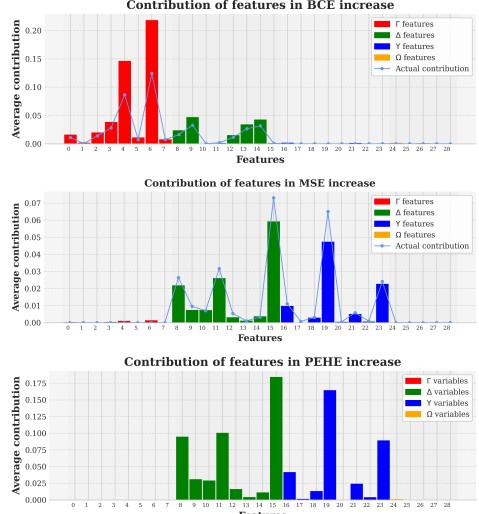

Figure 3: Feature permutation analysis demonstrating the inference and disentanglement of latent across BCE, MSE, and PEHE tasks.

and $\Delta$ features contribute to an increase in the BCE loss, validating our method's ability to accurately infer these factors within the data. Similarly, it confirms the successful learning of $\Delta$ and $\Upsilon$ factors, as evidenced by their impact on the MSE when permuted. Finally, it illustrates that GLOVE-ITE

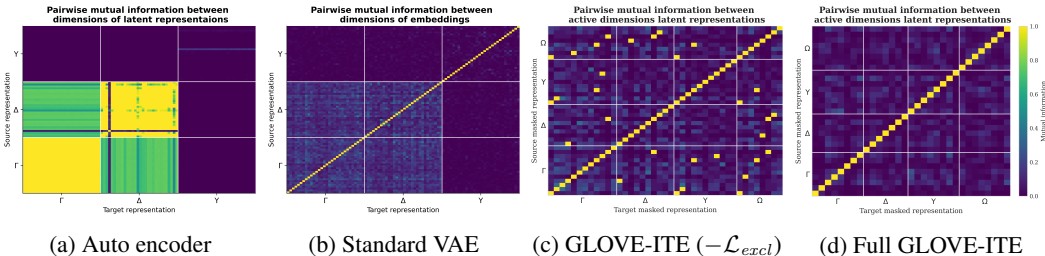

(a) Auto encoder     (b) Standard VAE     (c) GLOVE-ITE ($-\mathcal{L}_{excl}$)     (d) Full GLOVE-ITE

Figure 4: Pairwise mutual information heat maps: autoencoder and VAE show leakage, GLOVE-ITE without $\mathcal{L}_{excl}$ already reduces redundancy, and full GLOVE-ITE achieves disentangled, independent representations (dark = independence, light = leakage; best viewed in color).

effectively disentangles and infers $\Omega$, as permuting irrelevant features does not affect the PEHE, while permuting relevant features leads to a significant increase. This analysis underscores the robustness of GLOVE-ITE in inferring and disentangling the critical latent factors. Louizos et al. (2017); Löwe et al. (2022); Vowels et al. (2021) show that when latent factors are correctly inferred, causal effects can be identified and the reliance on the unconfoundedness assumption (see Appendix A.1.1) is reduced.

Figure 4 compares mutual information across latent representations for different approaches. Subfigure 4a shows that a simple autoencoder fails to separate factors, with substantial information leakage. Introducing VAEs (Subfigure 4b) reduces this leakage and improves disentanglement. Subfigure 4c demonstrates that GLOVE-ITE without $\mathcal{L}_{excl}$ already mitigates redundancy within and across representations, while the full GLOVE-ITE model (Subfigure 4d) achieves complete independence, yielding fully disentangled representations. Complementing this, Figure 5 visualizes the mechanism behind these results by showing the learned masks over four latent subspaces. Active dimensions are color-coded (red: $\Gamma$, green: $\Delta$, blue: $\Upsilon$, orange: $\Omega$), inactive ones are white, and overlaps (if any) appear in black. The exclusivity loss $\mathcal{L}_{excl}$ drives non-overlapping activations, ensuring each mask isolates only the dimensions relevant to its factor. This allocation prevents information leakage, reduces redundancy, and enhances interpretability.

**Tolerance ($\tau$) effect:** Figure 6 shows that MSE tolerance ($\tau$) inversely affects the number of active latent dimensions—higher $\tau$ yields fewer dimensions—while directly increasing PEHE. This trade-off underscores the need to carefully select $\tau$ to balance efficiency and estimation accuracy.

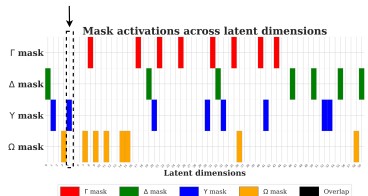

Figure 5: Learned mask matrix **S** with active dimensions (red: $\Gamma$, green: $\Delta$, blue: $\Upsilon$, orange: $\Omega$), inactive in white, and overlaps in black. The exclusivity loss $\mathcal{L}_{excl}$ enforces non-overlapping activations, and a dotted box marks an example of a dimension assigned to a single mask.

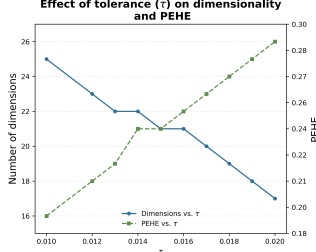

Figure 6: Active dimensionality and PEHE as functions of MSE tolerance.

## 6 CONCLUSION

In this work, we introduced a novel VAE-based framework that integrates the Generalized ELBO with Constrained Optimization (GECO) and an $L_0$ sparsity objective to automatically determine the effective latent dimensionality for treatment effect estimation. A mutual exclusivity regularizer further enforces independence across representations, strengthening disentanglement. By learning compact representations and isolating irrelevant variables, our method enhances accuracy, interpretability, and computational efficiency. Extensive experiments on real-world and synthetic datasets demonstrate that it consistently outperforms state-of-the-art baselines in predictive accuracy and robustness, particularly in high-dimensional settings with many irrelevant variables.

ETHICS STATEMENT

This work focuses on methodological contributions for treatment effect estimation and does not involve human subjects, sensitive data, or societal applications with direct ethical concerns. Therefore, we believe no specific ethical issues apply.

REPRODUCIBILITY STATEMENT

To ensure reproducibility, we provide:

- Complete descriptions of datasets in Section 5.2 and Appendix B.1.
- Detailed model architectures, hyperparameters, and optimization procedures in Appendix A.2 and B.3.
- Code, public and synthetic datasets, scripts, instructions for reproducing all experiments and to generate synthetic dateset are provided in supplementary material.

LLM USAGE

LLM is used just to polish writing.

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

# A    APPENDIX

## A.1    BACKGROUND KNOWLEDGE

In this section, we provide a cursory explanation of the Treatment Effect Estimation problem assumptions and the Variational Autoencoder (VAE) model.

### A.1.1    ASSUMPTIONS

Treatment effect estimation relies on the following widely adopted assumptions (Rosenbaum & Rubin., 1983; Rubin, 2005b; Imbens & Rubin, 2015).

- **Stable Unit Treatment Value:** The treatment assigned to one unit does not influence the potential outcomes of any other unit (i.e. patient).

- **Unconfoundedness:** There are no unmeasured confounders; all factors affecting both the treatment $\mathcal{T}$ and the outcome $\mathcal{Y}$ have been observed, formally, $\mathcal{Y} \perp\!\!\!\perp \mathcal{T} \mid \mathcal{X}$.

- **Overlap:** The probability of receiving any treatment given covariates $\mathbf{x}$ is strictly positive for all treatments.. Formally, $P(t \mid \mathbf{x}) > 0 \, \forall t \in \mathcal{T}, \forall \mathbf{x} \in \mathcal{X}$ (Rubin, 2005a).

- **Consistency:** Let $y^t$ denote the potential outcome under treatment $t \in \{0, 1\}$, and let $\mathcal{T}$ be the actual treatment received. The consistency assumption states that the observed outcome $y$ equals the potential outcome corresponding to the treatment actually received:

$$y = y^t \quad \text{if } \mathcal{T} = t$$

### A.1.2 Variational Autoencoders

Variational Autoencoders (VAEs) (Kingma & Welling, 2014) are deep generative models with a latent representation bottleneck. The generative process uses a prior distribution over the latent variables, $p(\mathbf{z})$ (commonly a Gaussian $\mathcal{N}(0, \mathbf{I})$, and a likelihood function $p(\mathbf{x} \mid \mathbf{z})$, parameterized by a neural network. The objective is to maximize the marginal likelihood $p(\mathbf{x}) = \int p(\mathbf{x}|\mathbf{z})p(\mathbf{z})d\mathbf{z}$, which is generally intractable. Instead, we optimize the Evidence Lower Bound (ELBO) on the marginal likelihood,

$$\mathcal{L}_{\text{ELBO}} = \mathbb{E}_{q(\mathbf{z}|\mathbf{x})}[\log p(\mathbf{x}|\mathbf{z})] - D_{\text{KL}}(q(\mathbf{z}|\mathbf{x})\|p(\mathbf{z})), \tag{12}$$

where $q(\mathbf{z}|\mathbf{x})$ is the learned variational posterior (encoder) that approximates the true posterior $p(\mathbf{z}|\mathbf{x})$. The reconstruction loss in the expectation ensures that the latent representation preserves sufficient information and the Kullback-Leibler (KL) divergence induces a structured and smooth latent space. The encoder maps $\mathbf{x}$ to the parameters of a Gaussian

$$q(\mathbf{z}|\mathbf{x}) = \mathcal{N}(\boldsymbol{\mu}(\mathbf{x}), \text{diag}(\boldsymbol{\sigma}^2(\mathbf{x}))), \tag{13}$$

while the decoder reconstructs $\mathbf{x}$ from sampled $\mathbf{z}$.

### A.2 Experiment Details

IHDP Dataset: We utilized a single representational network with three layers and an input of 60 dimensions. The hidden and output layers contained 100 and 60 neurons respectively. The network was trained using the Adam optimizer with ELU activation, a batch size of 500, and a learning rate of $1e-5$ over a maximum of 7000 epochs. The best model was selected using PEHE on the validation set, following the approach in Shalit et al. (2017). Data splitting for training, validation, and testing matched the protocol with 20% of the training data reserved for validation. For GECO, we set $\lambda_{min} = 0$, $\lambda_{max} = 200$, $\lambda_{init} = 1$, $\alpha = 0.99$, and an MSE tolerance ($\tau$) of 0.4. For $L_0$ regularization, we used $beta = 0.6$, $\gamma = -0.1$, $\zeta = 1.1$ and $\kappa = 0.05$.

Synthetic Dataset: The same settings were applied with the following modifications: input and output dimensions for the encoder were 60, batch size was 512, maximum epochs were limited to 1500, MSE tolerance was set to 0.01 and $\kappa = 0.05$.

## B Jobs Experiments

### B.1 Dataset

Jobs is an observational dataset based on the Lalonde experiment (LaLonde, 1986), containing eight pre-treatment covariates (age, educ, black, hisp, married, nodegr, re74, re75). The binary treatment indicates participation in job training, while the outcome is post-training earnings in 1978. The dataset has 614 instances (185 treated, 429 control) (Hill, 2011; Dehejia & Wahba, 1999), and we use artificial contrasts for evaluation.

### B.2 Policy Risk: evaluation criteria

Policy Risk ($\mathcal{R}_{pol}$) is defined as:

$$\mathcal{R}_{pol}(\pi_f) = 1 - \left(\mathbb{E}[Y^1 \mid \pi_f(x) = 1]\, p(\pi_f = 1) + \mathbb{E}[Y^0 \mid \pi_f(x) = 0]\, p(\pi_f = 0)\right), \tag{14}$$

where the treatment policy $\pi_f(x)$ is induced by the model $f$. Specifically, $\pi_f(x) = 1$ (treat) if $\hat{y}^1 - \hat{y}^0 > \lambda$ and $\pi_f(x) = 0$ otherwise. This criterion measures the expected loss in value of following $\pi_f$, with expectations weighted by the probability of treatment assignment (Shalit et al., 2017).

### B.3 Experiment details

For the Jobs dataset, we followed the same experimental protocol as for IHDP, except with an MSE tolerance of 0.3 and a training budget of 3000 epochs. The representational network had three layers with input dimension 60, hidden layer size 100, and output size 60, trained with Adam (learning rate

$1e-5$, batch size 500, ELU activations). The best model was selected using validation $\mathcal{R}_{pol}$ (Shalit et al., 2017), with $20\%$ of training data held out for validation. For GECO, we used $\lambda_{min} = 0$, $\lambda_{max} = 200$, $\lambda_{init} = 1$, $\alpha = 0.99$, and $\tau = 0.4$; for $L_0$ regularization, $\beta = 0.6$, $\gamma = -0.1$, $\zeta = 1.1$, and $\kappa = 0.05$.

## B.4 RESULTS

Table 4: $\mathcal{R}_{pol}$ (mean(std)) on Jobs dataset under varying numbers of irrelevant variables ($\#\Omega$)– Lower is better. Reported values include active latent dimensionalities per method.

| Method | $\#\Omega$ | $\Gamma$-active | $\Delta$-active | $\Upsilon$-active | $\Omega$-active | Total-active | $\mathcal{R}_{pol}$ |
|---|---|---|---|---|---|---|---|
| DR-CFR | 5 | 15 | 15 | 15 | NA | 45 | 0.18 (0.06) |
| TEDVAE | 5 | 15 | 15 | 15 | NA | 45 | 0.20 (0.03) |
| TVAE | 5 | 15 | 15 | 15 | 15 | 60 | 0.14 (0.01) |
| DRI-ITE | 5 | 15 | 15 | 15 | 15 | 60 | 0.16 (0.05) |
| **GLOVE-ITE** | 5 | 8 | 9 | 7 | 8 | 32 | **0.13 (0.006)** |
| DR-CFR | 15 | 15 | 15 | 15 | NA | 45 | 0.17 (0.06) |
| TEDVAE | 15 | 15 | 15 | 15 | NA | 45 | 0.21 (0.04) |
| TVAE | 15 | 15 | 15 | 15 | 15 | 60 | 0.15 (0.01) |
| DRI-ITE | 15 | 15 | 15 | 15 | 15 | 60 | 0.14 (0.03) |
| **GLOVE-ITE** | 15 | 8 | 9 | 7 | 8 | 32 | **0.13 (0.006)** |
| DR-CFR | 20 | 15 | 15 | 15 | NA | 45 | 0.17 (0.06) |
| TEDVAE | 20 | 15 | 15 | 15 | NA | 45 | 0.19 (0.03) |
| TVAE | 20 | 15 | 15 | 15 | 15 | 60 | 0.22 (0.08) |
| DRI-ITE | 20 | 15 | 15 | 15 | 15 | 60 | 0.14 (0.03) |
| **GLOVE-ITE** | 20 | 8 | 9 | 7 | 8 | 32 | **0.13 (0.006)** |

On the Jobs dataset (Table 4), competing VAE-based methods either retain the full latent dimensionality or fail to disentangle irrelevant variables, leading to consistently higher policy risk across all settings. For example, TVAE and DRI-ITE keep 60 active dimensions and still yield higher $\mathcal{R}_{pol}$ values, while TEDVAE underperforms despite fewer dimensions. In contrast, GLOVE-ITE maintains a compact representation (about 32 active dimensions) and achieves the lowest policy risk (0.13) across all cases. These results confirm that our method simultaneously reduces dimensionality, isolates irrelevant variables, and improves policy robustness by preventing spurious dependencies.

