# OpenReview forum: "Sparse Disentangled VAE for Treatment Effect Estimation with Irrelevant Variables"
_ICLR.cc/2026/Conference — ICLR 2026 Conference Withdrawn Submission_

### Official Review · Reviewer_C5JF · 2025-10-25

**Soundness:** 3
**Presentation:** 2
**Contribution:** 3
**Rating:** 6
**Confidence:** 3

**Summary:**

This paper introduces GLOVE-ITE, a novel VAE-based framework for treatment effect estimation from high-dimensional observational data where irrelevant variables are present. The paper's core contribution is tackling a key limitation of existing VAE methods: the reliance on a pre-defined, fixed latent bottleneck width. The authors argue this leads to under- or over-fitting, as the dimensionality is critical but impractical to cross-validate.

GLOVE-ITE automatically learns a sparse, disentangled representation by integrating three key techniques:

1. Differentiable $L_0$ Sparsity: It employs $L_0$ regularization with a Binary Concrete relaxation to learn sparse, task-specific masks for different latent factors (instrumental $\Gamma$, confounding $\Delta$, adjustment $\Upsilon$, and irrelevant $\Omega$). This effectively performs automatic dimension selection.

2. GECO (Generalized ELBO with Constrained Optimization): To balance the conflicting objectives of prediction accuracy (MSE) and sparsity ($L_0$), the method uses GECO. This reframes the outcome prediction loss as a constraint, ensuring that sparsity is only optimized once the prediction error drops below a specified tolerance threshold $\tau$.

3. Exclusivity Regularizer ($\mathcal{L}_{excl}$): A novel entropy-based loss is introduced to enforce that the learned masks are mutually exclusive (non-overlapping). This prevents information leakage between factors and promotes stronger disentanglement, particularly in isolating the irrelevant variables $\Omega$.

The authors conduct extensive experiments on synthetic data and two real-world benchmarks. The results demonstrate that GLOVE-ITE achieves state-of-the-art or competitive accuracy while using a significantly more compact latent representation.

**Strengths:**

- **Novel and Sound Methodology**: The primary strength is the elegant methodological design. The use of GECO to dynamically balance the conflicting accuracy and sparsity objectives is a key innovation that moves beyond simple weighted losses. This is powerfully supported by the $\mathcal{L}_{excl}$ loss, which ensures the learned sparse dimensions are also meaningfully disentangled.

- **Solves a Critical Practical Problem**: The paper successfully automates the selection of latent dimensionality, a notoriously difficult hyperparameter to tune. The compression analysis in Table 3 provides strong evidence that the model can discover the "true" intrinsic dimension of the data.

**Weaknesses:**

- **Sensitivity to GECO's $\tau$ Hyperparameter**: The paper elegantly removes the need to tune the latent dimensionality, but it introduces a new, and potentially equally sensitive, hyperparameter: the MSE tolerance $\tau$ for GECO. Figure 6 explicitly shows that both the final PEHE and the number of active dimensions are highly dependent on the choice of $\tau$. The paper provides the $\tau$ values used, but does not discuss how these values were chosen, which risks trading one difficult hyperparameter for another.

- The study primarily relies on two small-scale real-world datasets and one synthetic dataset. Although the paper claims its method has advantages in "high-dimensional data" and "large data settings," the real-world datasets used have very small sample sizes. This weakens the conclusions regarding the method's scalability and robustness on large-scale problems.

**Questions:**

- Could you elaborate on the process for selecting the GECO tolerance hyperparameter $\tau$? Figure 6 suggests performance is quite sensitive to it. How were the values of 0.4 (for IHDP/Jobs) and 0.01 (for Synthetic) determined? Is there a principled way to set $\tau$ that avoids the same level of exhaustive tuning that the paper sought to eliminate for bottleneck width?

- The paper claims to address the issue of "predetermined latent dimensionality" and "eliminate the need for predefined or manually tuned bottleneck width." However, in practice (as shown in Table 3), the model still requires the user to specify an initial total latent dimension $d$ (e.g., 60, 68, 76, 84). GLOVE-ITE learns a sparse active subset from this initial $d$-dimensional space, rather than determining the dimensionality from scratch. This initial $d$ remains a critical hyperparameter that must be defined by the user. The paper does not discuss how the model’s performance might be affected if this initial $d$ is set improperly (e.g., if it is smaller than the true intrinsic dimensionality)..

---

### Official Review · Reviewer_ukKR · 2025-10-26

**Soundness:** 2
**Presentation:** 2
**Contribution:** 2
**Rating:** 2
**Confidence:** 4

**Summary:**

This paper proposes GLOVE-ITE, a VAE-based framework designed to estimate individual treatment effects in high-dimensional observational data that may contain irrelevant or confounding variables. The authors claim that this framework allows the model to automatically determine the optimal latent dimensionality and improve interpretability while maintaining competitive prediction performance. Experiments on synthetic, IHDP, and Jobs datasets show modest improvements in PEHE and policy risk over baselines.

**Strengths:**

1. Addressing the impact of irrelevant variables in high-dimensional ITE estimation is a relevant and practically important topic.
2. The experiments follow standard benchmarks, and the results indicate some improvement over prior VAE-based baselines.

**Weaknesses:**

1. The main components of the proposed method are well-established techniques. Their combination is incremental and lacks clear conceptual innovation. There is no new causal modeling insight beyond architectural tuning.
2. The paper claims to address causal disentanglement, but the proposed method operates purely at the representational level without formal causal guarantees.
3. The use of GECO and L0 regularization is motivated empirically but not theoretically linked to causal identifiability or bias reduction. The mathematical formulations are standard VAE objectives with additional penalty terms, not fundamentally new learning principles.
4. The experiments are limited to small-scale datasets (IHDP, Jobs) and synthetic examples. There is no evidence that the method scales to realistic high-dimensional or nonlinear causal systems.
5. The visualization and discussion of “latent causal factors” are not convincing. The masks and sparsity patterns are not compared against ground-truth causal variables, making it unclear whether the model truly learns meaningful disentanglement.

**Questions:**

See Weaknesses Part.

---

### Official Review · Reviewer_LaRH · 2025-10-28

**Soundness:** 2
**Presentation:** 1
**Contribution:** 1
**Rating:** 2
**Confidence:** 4

**Summary:**

The paper introduces a variational autoencoder-based approach for estimating treatment effects from observational data, focusing on disentangling latent factors to improve causal inference. Key components include:

- **Learnable Masks via L0 Regularization**: Binary masks are used to partition the latent space into four distinct groups—instrumental variables, confounding factors, adjustment variables, and irrelevant dimensions—allowing selective access to relevant latents for different modeling tasks (e.g., treatment assignment and outcome prediction).

- **Generalized ELBO with Constrained Optimization (GECO)**: to trade off reconstruction fidelity, outcome prediction accuracy, and sparsity in the masks.

- **Entropy Regularizer for Mutual Exclusivity**: A regularization term that penalizes overlap among the masks by promoting high entropy in their assignments, thereby enforcing separation of the latent groups and reducing information leakage between causal factors.

**Strengths:**

- The integration of these components—masks for latent partitioning, GECO for multi-objective optimization, and entropy-based separation—appears novel in its specific application to causal disentanglement for treatment effect estimation.
- Experiments demonstrate empirical gains, including improved Precision in Estimation of Heterogeneous Effects (PEHE) scores and visualizations of well-separated latent representations across groups.

**Weaknesses:**

### Unidentifiable Representations

The core issue lies in the unidentifiability of the learned representations, especially the mask-based separation. Identifiable representations are tied to the data(-generating process) and remain invariant to training procedures (see [1] for a detailed framework). Here, even with the pre-trained "full representation" fixed, the learnable masks yield inconsistent outputs. For instance:

- Varying the hyperparameter $\lambda$ produces different mask matrices,
- Changing the random seed during training likely results in varied masks.

However, the underlying causal graph of covariates being unique (or equivalent up to Markov equivalence classes for treatment effect estimation).
This means the method fails to recover the true causal structure reliably.

### Potential Dependence in Selected Dimensions

Even if masks select dimensions, dependence can persist if the pre-separated "full representations" are not independent. The resulting instrumental, confounding, and adjustment latents may not be fully disentangled. To achieve independence, an identifiable VAE (iVAE) framework [1] is necessary, with direct applications to PEHE estimation available in [2].

### Limited Novelty

The primary ideas are taken (or straightforward adaptations) from prior work:

- Binary Concrete distributions for implementing sparse masks.
- GECO as a regularization to penalize outcome prediction errors.
- Partitioning latents into instrumental, confounding, and adjustment categories.
- L0 regularization via masks reduces to selecting latent dimensions for causal roles.

### Arbitrary Method Design Choices

Several decisions seem ad hoc or suboptimal:

- **Irrelevant Representations**: Why introduce explicit masks for irrelevant dimensions? Any dimension not assigned to the three causal masks (instrumental, confounding, adjustment) is inherently irrelevant, making this redundant.
- **GECO vs. Simpler Alternatives**: Instead of GECO, one could multiply the mask loss by a hyperparameter to balance sparsity and prediction error—simpler and equally effective. Alternatively, incorporating an outcome likelihood term (analogous to those for covariates $X$ and treatment $T$) would more naturally integrate prediction without custom constraints.
- **Dependence on $\lambda$**: The method relies on tuning the error threshold $\lambda$ for sparsity, akin to manually selecting latent dimensionality as a bottleneck—both are heuristic ways to control model capacity without deeper justification.


### Experiments

- **Table 1**: Standard deviations for baseline methods are unusually large, suggesting a potential bug.
- **Figure 6**: I think PEHE accuracy should degrade for very small $\lambda$ due to overfitting from insufficient sparsity.


### Writing Issues

The writing is often unclear, imprecise, or repetitive. Specific examples:

- **Abstract**: "Treatment effect estimation from imbalanced observational data is challenging…" (First Sentence) and later parts of the paper mention balancing or balanced representations repeatedly, but this is not a novel contribution and creates confusion about the paper's focus.
- **Introduction**: Vague and redundant phrases like "entanglement between distinct latent causal factors interferes with the model’s causal structure" and "information leakage between latent factors can lead to inaccurate inference of latent factors." Terms such as "dedicated mask that separates irrelevant factors into a distinct latent subspace" and "mutual exclusivity regularization across all masks that prevents information leakage" are imprecise without later context.
- **Binary Concrete Distribution (for Masks)**: Unclear details—e.g., are $\alpha$ and $\beta$ hyperparameters? How does $\alpha$ influence the process? The transition from Equation (2) to (3) lacks explanation, and the rationale for the "stretching" equation is absent.
- **Line 234**: "Other tasks perform better than a threshold" uses plural "tasks" (likely meaning outcome prediction), which is confusing.
- **Equation Ordering**: Presenting incomplete Equation (4) before (9), while referencing (9) early (Line 248), disrupts logical flow.
- **Optimization Details**: The min-max optimization scheme is undefined—e.g., does it have a stochastic gradient version? Which optimizer was used?

### Missing Related Work

The paper omits key references on identifiable representations and causal VAEs:

- [1] Khemakhem, Ilyes, et al. "Variational autoencoders and nonlinear ICA: A unifying framework." *International Conference on Artificial Intelligence and Statistics*. PMLR, 2020. (For foundational identifiability in VAEs.)
- [2] Wu, Pengzhou Abel, and Kenji Fukumizu. "$\beta$-Intact-VAE: Identifying and Estimating Causal Effects under Limited Overlap." *International Conference on Learning Representations* (2022). (For iVAE applications to PEHE and disentanglement.)

**Questions:**

Please refer to the points in Weaknesses.

---

### Official Review · Reviewer_JcSA · 2025-10-30

**Soundness:** 3
**Presentation:** 3
**Contribution:** 2
**Rating:** 4
**Confidence:** 4

**Summary:**

The paper proposes GLOVE-ITE, a VAE-based framework for individual treatment effect (ITE) estimation that learns a compact, disentangled latent space in the presence of irrelevant variables. It adopts the L0 sparsity objective that learns masks over a shared latent to activate only task-relevant dimensions and utilizes GECO to prioritize outcome prediction while sparsifying. Experiments on IHDP, Jobs, and a structured synthetic dataset show better performance with fewer active latent dims than VAE baselines, supported by ablations and qualitative analyses.

**Strengths:**

- The problem, underlying factors, and losses are clearly laid out with equations and a helpful architecture figure.
- Automatic bottleneck sizing approach in this context is interesting.
- Claims are supported by extensive experimental results.
-  Key hyperparameters and datasets are sufficiently described.

**Weaknesses:**

- Novelty appears to be incremental. Prior work combined GECO + sparsity in VAEs; a clearer positioning vs. (Boom et al. 2020) is needed.
- Baselines are VAE-centric; adding strong non-VAE ITE methods (e.g., modern CFR variants or representation learners without generative modeling) would better calibrate performance gains.
- While (\tau) effects are shown, broader sweeps for $\beta$, $\gamma$ , $\zeta$, and $\kappa$ are missing.

**Questions:**

1. Beyond the target domain and exclusivity loss, what prevents (Boom et al. 2020)-style approaches from achieving similar behavior? Can you ablate exclusivity against simple orthogonality penalties?
2. Please include non-VAE ITE SOTA baselines (e.g., strong CFR/DR learners).
3. Please report sensitivity to ($\beta,\gamma,\zeta,\kappa$); provide recommended ranges to avoid degenerate solutions.

---

### Note · Authors · 2025-11-19

I have read and agree with the venue's withdrawal policy on behalf of myself and my co-authors.